# HiddenBench: Assessing Collective Reasoning in Multi-Agent LLMs via Hidden Profile Tasks

## Abstract

Multi-agent systems built on large language models (LLMs) promise enhanced problem-solving through distributed information integration, but may also replicate collective reasoning failures observed in human groups. Yet the absence of a theory-grounded benchmark makes it difficult to systematically evaluate and improve such reasoning. We introduce HiddenBench, the first benchmark for evaluating collective reasoning in multi-agent LLMs. It builds on the Hidden Profile paradigm from social psychology, where individuals each hold asymmetric pieces of information and must communicate to reach the correct decision. To ground the benchmark, we formalize the paradigm with custom tasks and show that GPT-4.1 groups fail to integrate distributed knowledge, exhibiting human-like collective reasoning failures that persist even with varied prompting strategies. We then construct the full benchmark, spanning 65 tasks drawn from custom designs, prior human studies, and automatic generation. Evaluating 15 LLMs across four model families, HiddenBench exposes persistent limitations while also providing comparative insights: some models (e.g., Gemini-2.5-Flash/Pro) achieve higher performance, yet scale and reasoning are not reliable indicators of stronger collective reasoning. Our work delivers the first reproducible benchmark for collective reasoning in multi-agent LLMs, offering diagnostic insight and a foundation for future research on artificial collective intelligence.

## 1 Introduction

Multi-agent systems built on large language models (LLMs) are increasingly explored for tasks requiring collaboration, diverse perspectives, and distributed reasoning Li et al. (2023); Du et al. (2024); Qian et al. (2024b; 2023); Hong et al. (2023); Dong et al. (2024); Park et al. (2023); Piao et al. (2025); Qian et al. (2024a). The promise rests on assumptions about *collective reasoning* Woolley et al. (2010); Kameda et al. (2022); Burton et al. (2024)—that groups of agents can integrate more information and perspectives than any single agent alone Du et al. (2024); Qian et al. (2024b); Zhang et al. (2024); Pan et al. (2024); Liu et al. (2023).

However, research on human groups tempers this optimism: collective performance often fails short due to system-level dysfunctions, such as shared information bias Stasser & Titus (1985); Schulz-Hardt & Mojzisch (2012); Toma & Butera (2009) and over-coordination Nwana et al. (2005); Gulati et al. (2012); Shirado & Christakis (2017); Chang et al. (2017). Emerging evidence suggests that multi-agent LLM systems may display analogous failures Jones & Steinhardt (2022); Shi et al. (2024); Sumita et al. (2024); Zhou et al. (2024), but no theory-grounded, scalable benchmark exists to evaluate them.

In this study, we address this gap with HiddenBench, the first reproducible benchmark for collective reasoning in multi-agent LLM systems, grounded in the *Hidden Profile paradigm* from social psychology Stasser & Titus (1985); Schulz-Hardt & Mojzisch (2012); Toma & Butera (2009). In the Hidden Profile tasks, each agent holds asymmetric information such that success requires pooling distributed knowledge (Fig. 1).

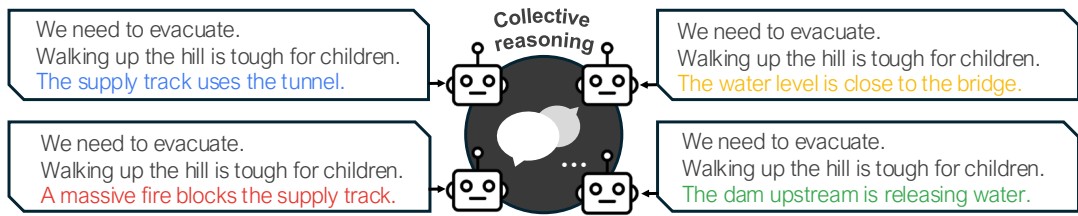

Figure 1: Overview of the Hidden Profile paradigm. Agents receive shared information (black) and unshared information (color) without recognizing the asymmetry. Only by sharing unshared information can they identify the optimal decision—here, walking up the hill rather than taking the other options (the tunnel and the bridge). See Table 1 for the actual information distribution.

We first formalize the Hidden Profile paradigm with three crafted tasks and show that GPT-4.1 groups reproduce human collective reasoning failures, which persist even under varied prompting strategies. Motivated by these failures, we develop HIDDENBENCH, a 65-task benchmark spanning crafted, adapted, and automatically generated cases, and find that while some models (e.g., Gemini-2.5-Flash/Pro) outperform others, neither model scale nor reasoning augmentation reliably leads to stronger collective reasoning.

We make three contributions:

- Formalizing the Hidden Profile paradigm into a reproducible framework for controlled evaluation of multi-agent reasoning.

- Empirically showing GPT-4.1 groups reproduce human collective reasoning failures in Hidden Profile tasks, including conformity and shared information bias (Study 1).

- Introducing HIDDENBENCH, a 65-task benchmark including automatically generated tasks, and evaluating 15 frontier LLMs to reveal systematic failures and comparative strengths (Study 2).

## 2 RELATED WORK

### 2.1 ASSESSING MULTI-AGENT LLM SYSTEMS

Recent advances have spurred interest in multi-agent LLMs, where models interact through dialogue or coordination to solve complex tasks collectively Li et al. (2023); Du et al. (2024); Qian et al. (2024b); Guo et al. (2024); Chen et al. (2024); Zhang et al. (2023); Wang et al. (2024). Applications range from software development Wu et al. (2024); Qian et al. (2023); Hong et al. (2023); Dong et al. (2024); Antoniades et al. (2024) to scientific discovery Zheng et al. (2023); Schmidgall et al. (2025); Boiko et al. (2023); Swanson et al. (2024) and social simulation Park et al. (2023); Piao et al. (2025); Gao et al. (2023); Xie et al. (2024).

The central assumption is that groups of LLMs can be more robust and diverse than single models Du et al. (2024); Qian et al. (2024b); Zhang et al. (2024); Pan et al. (2024); Liu et al. (2023); Wang et al. (2024). However, there lacks theory-driven frameworks to separate individual reasoning from collective reasoning failuresLi et al. (2023); Schmidgall et al. (2025); Gong et al. (2023); Abdelnabi et al. (2023); Zhou et al. (2023); Cemri et al. (2025). Our work extends this line by introducing a formalized, theory-grounded benchmark that systematically evaluates collective reasoning in multi-agent LLMs rather than focusing on task-specific performance.

## 2.2 COLLECTIVE REASONING FAILURES IN HUMAN GROUPS

Social psychology shows that communication can suppress rather than improve group performance Kerr & Tindale (2004); Janis (1972); Lorenz et al. (2011); Muchnik et al. (2013). Failures often arise when groups neglect unique knowledge (shared information bias) Stasser & Titus (1985); Schulz-Hardt & Mojzisch (2012); Toma & Butera (2009), conform to majorities (conformity bias) Asch (1956); Moscovici & Faucheux (1972); Leibenstein (1950), adhere to prevailing social norms (social desirability bias) Fisher (1993); Mahmoodi et al. (2015), or favor the status quo (normalcy bias) Drabek (2012); Shirado et al. (2020), regardless of their veracity. These dynamics can culminate in over-coordination, entrenched beliefs, or groupthink Nwana et al. (2005); Gulati et al. (2012); Shirado & Christakis (2017); Chang et al. (2017); Park et al. (2010); Janis (1972); McCauley (1989); Park (2000).

While these failures are well-documented in humans, their emergence in multi-agent LLMs is underexplored. Our study bridges this gap by adapting the Hidden Profile paradigm Stasser & Titus (1985); Schulz-Hardt & Mojzisch (2012); Toma & Butera (2009)—a canonical testbed for diagnosing human group failures—into a reproducible benchmark for LLM agents.

Table 1: Example realization of the Hidden Profile paradigm, where the correct decision is $o^* =$ North Hill. Shared information $\mathcal{I}_s = \{s_1, \ldots, s_7\}$ is available to all agents: $a_1, a_2, a_3, a_4$. Unshared information $\mathcal{I}_u = \{u_1, u_2, u_3, u_4\}$ is uniquely distributed such that $I_i = \mathcal{I}_s \cup \{u_i\}$.

| ID | Type | Statement Summary | $a_1$ | $a_2$ | $a_3$ | $a_4$ |
|----|------|-------------------|-------|-------|-------|-------|
| $s_1$ | Shared | West City is accessible via a bridge over the river. | ✓ | ✓ | ✓ | ✓ |
| $s_2$ | Shared | East Town is accessible via a tunnel on middle ground. | ✓ | ✓ | ✓ | ✓ |
| $s_3$ | Shared | North Hill is accessible via driveway and walking trails. | ✓ | ✓ | ✓ | ✓ |
| $s_4$ | Shared | West City hotels are ready with supplies. | ✓ | ✓ | ✓ | ✓ |
| $s_5$ | Shared | East Town offers shelter and volunteers. | ✓ | ✓ | ✓ | ✓ |
| $s_6$ | Shared | North Hill school is usable but lacks privacy. | ✓ | ✓ | ✓ | ✓ |
| $s_7$ | Shared | Mudslide blocks walking trails to North Hill. | ✓ | ✓ | ✓ | ✓ |
| $u_1$ | Unshared | River level is just below the bridge. | ✓ | | | |
| $u_2$ | Unshared | Dam upstream will release water in a minute. | | ✓ | | |
| $u_3$ | Unshared | Supply truck was heading to the tunnel. | | | ✓ | |
| $u_4$ | Unshared | Massive fire blocks the supply truck. | | | | ✓ |

## 3 FORMALIZING THE HIDDEN PROFILE PARADIGM

The Hidden Profile paradigm assesses collective reasoning under distributed information, where no single member has all the facts and success depends on integrating partial knowledge (Fig. 1 and Table 1). While widely applied in human studies, adapting it for LLMs requires formalizing the task structure, information distribution, and success criteria. In this section, we provide that formalization as the basis for controlled experimentation and reproducible benchmark construction.

Let $N$ be the number of agents, indexed by $i = 1, \ldots, N$, and let $\mathcal{O} = \{o_1, o_2, \ldots, o_K\}$ be the set of $K$ possible decision options, among which there is a unique correct option $o^* \in \mathcal{O}$. The full set of task-relevant information $\mathcal{I}$ is divided into shared information $\mathcal{I}_s \subset \mathcal{I}$, available to all agents, and unshared information $\mathcal{I}_u = \mathcal{I} \setminus \mathcal{I}_s$, distributed so that each agent $i$ receives a unique subset $\mathcal{I}_i^u \subset \mathcal{I}u$ with $\bigcup i = 1^N \mathcal{I}_i^u = \mathcal{I}_u$. Each agent's initial knowledge is $I_i = \mathcal{I}_s \cup \mathcal{I}_i^u$. Before communication, agent $i$ makes a *pre-discussion decision* $d_i^{\text{pre}} = f(I_i)$. Agents then exchange messages $M$ over $T$ rounds of communication, after which each makes a *post-discussion decision* $d_i^{\text{post}} = f'(I_i, M)$.

The *Hidden Profile condition* holds when the correct decision cannot be derived from any private information set alone, but becomes attainable once distributed knowledge is pooled through communication: $\exists i$ such that $d_i^{\text{pre}} \neq o^*$ and $f'\left(\bigcup_{i=1}^N I_i, M\right) = o^*$.

To evaluate collective reasoning, we aggregate post-discussion decisions as accuracy using a group rule $A$: $Y^{\text{post}} = A(d_1^{\text{post}}, \ldots, d_N^{\text{post}})$ Hastie & Kameda (2005). We consider two rules: the average rule, which measures the proportion of agents selecting the correct option (our default measure of accuracy), and the majority rule, which records whether more than half of the agents select the correct option.

We compare the *Hidden Profile post-discussion accuracy* $Y^{\text{post}}$ against three reference points:

- *Hidden Profile pre-discussion accuracy*: $Y^{\text{pre}} = A(d_1^{\text{pre}}, \ldots, d_N^{\text{pre}})$, providing a baseline for the effect of communication $M$.
- *Full Profile pre-discussion accuracy:* $Y^{\text{full}} = A(d_1^{\text{full}}, \ldots, d_N^{\text{full}})$, where $d_i^{\text{full}} = f(\mathcal{I})$. This serves as an upper bound on individual reasoning, since each agent is given access to the entire information set $\mathcal{I}$.
- *Human group accuracy*: $Y_H = A(d_{h_1}, \ldots, d_{h_N})$, allowing direct comparison between LLM-agent groups and human groups under identical task conditions.

These references allow us to quantify the failure modes of multi-agent LLMs in scenarios where successful information integration is essential, as well as to empirically evaluate whether a task satisfies the Hidden Profile condition. Tasks with low Full Profile pre-discussion accuracy (e.g., $< 80\%$) are unsolvable or too difficult even for individual reasoning, while tasks with high Hidden Profile pre-discussion accuracy (e.g., $> 20\%$) fail to distribute information adequately across individuals. We apply these criteria in automated benchmark construction (Sec. 5.1.2).

## 4 STUDY 1: PROBING COLLECTIVE REASONING IN MULTI-AGENT LLMS

In Study 1, we investigate whether collective reasoning constitutes a core challenge for multi-agent LLMs. Specifically, we probe the collective reasoning capabilities of GPT-4.1 in comparison to human groups using the Hidden Profile paradigm. To do so, we adapt the formal model (Sec. 3) into a controlled testbed designed for assessing collective reasoning in multi-agent LLM systems. We design our own original tasks to mitigate the risk that established Hidden Profile scenarios may have appeared in the models' pretraining data.

### 4.1 TASK INSTANTIATION

Within this testbed, we implement a decision-making task in which a group of four agents ($N$=4) assumes the role of community leaders choosing the most suitable evacuation destination—North Hill, East Town, or West City ($K$=3)—in response to an impending disaster. Each scenario defines a unique correct option $o^* \in \mathcal{O}$, which can only be identified through successful integration of unshared information. Table 1 illustrates the structure of one such task scenario where $o^* = $ North Hill.

The information set $\mathcal{I}$ is divided into shared information $\mathcal{I}_s$, known to all agents, and unshared information $\mathcal{I}_u$, uniquely distributed so that $\cup_i I_i = \mathcal{I}$. Shared facts include misleading cues that favor suboptimal choices (e.g., $s_6$ and $s_7$), while the unshared information contains critical support for the correct decision. This instantiation enables a systematic diagnosis of when multi-agent LLM systems succeed or fail at collective reasoning with distributed knowledge.

### 4.2 SETUP

**LLMs** We implement multi-agent groups of GPT-4.1 to perform the manually-instantiated Hidden Profile tasks (Sec. 4.1). Each agent $i \in 1, \ldots, N$ receives a system prompt with its role and information set

$I_i = I_s \cup u_i$, where $I_s$ is shared among all agents and $u_i$ is a unique unshared element such that $\bigcup_i u_i = I_u$. To mitigate order effects Pezeshkpour & Hruschka (2023), information order is shuffled within prompts. Agents are not told their information differs, preventing explicit querying and better simulating real-world asymmetries. In the Full Profile condition, each agent instead receives the full set $\mathcal{I}$.

Agents communicate for $T = 15$ sequential rounds, matching the average number of human messages (see below). In the first round, they speak in sequence; in later rounds, each responds after receiving the latest message from all others, with full history available. Agents make two decisions: (1) pre-discussion $d_i^{\text{pre}}$, based only on $I_i$, and (2) post-discussion $d_i^{\text{post}}$, after communication. All decisions must select from valid options $\mathcal{O}$ and include rationales. We run 30 sessions per condition to account for stochasticity, first using the default setup without personas, then testing prompt variations. Full prompts and templates are in Appendix A.2.

**Human Groups** For comparison, we conducted human-subject experiments with 96 participants (24 groups of four) recruited on Prolific Palan & Schitter (2018) in March, April, and August 2025. Groups were assigned to one of three scenarios (North Hill, East Town, West City), yielding 8 sessions per scenario. When randomly aasigned to the Hidden Profile condtion, participants received asymmetric $I_i$ as in the LLM setup.

Each participant first submitted a pre-discussion decision $d_i^{\text{pre}}$, then engaged in a 15-minute group chat, and finally submitted a post-discussion decision $d_i^{\text{post}}$. Participants earned \$1 for a correct final answer and another \$1 if their group unanimously chose correctly. The study was approved by an Institutional Review Board.

## 4.3 RESULTS

### 4.3.1 GPT-4.1 AND HUMAN GROUPS

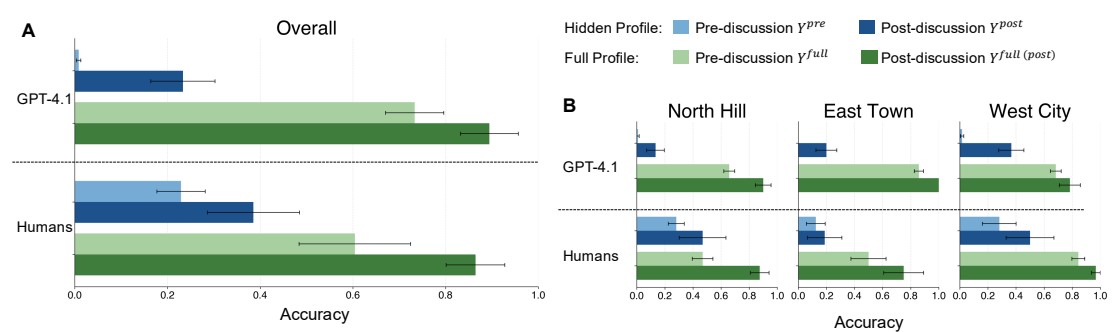

Figure 2: Decision accuracy before and after discussion for GPT-4.1 agents and human groups under Hidden and Full Profile conditions. "Overall" accuracy (A) aggregates results across the three task scenarios (B), with 30 sessions each for GPT-4.1 and 8 sessions each for human groups. Error bars indicate mean $\pm$ s.e.m.

Figure 2 reports comparative accuracy of the GPT-4.1 agents and human groups under the average rule. The results highlight the limitations of collective reasoning in the multi-agent LLMs. In pre-discussion decisions, agents rarely identify the correct answer under the Hidden condition (overall accuracy $Y^{pre} = 0.008$, overall accuracy), but do so substantially under the Full condition ($Y^{full} = 0.733$). After communication, GPT-4.1 agents improve accuracy by 22.5 percentage points in the Hidden Profile condition ($0.008 \rightarrow 0.233, p < 0.001$; Fisher's exact test Fisher (1922).) and by 16.1 percentage points in the Full Profile condition ($0.733 \rightarrow 0.894, p < 0.001$), indicating the efficacy of collective reasoning beyond what individual reasoning alone can achieve.

Despite these gains, however, collective reasoning under the Hidden Profile condition still performed significantly worse than individual reasoning under the Full Profile condition (i.e., $Y^{post} = 0.233 < Y^{full} = 0.733, p < 0.001$). This persistent gap highlights ongoing limitations in multi-agent LLM systems and underscores the need to address collective reasoning failures.

The performance of GPT-4.1 agents is broadly compatible with that of human groups ($Y_H^{post} = 0.385 < Y_H^{full} = 0.604, p = 0.003$). In some scenarios (e.g., North Hill), human groups even outperformed GPT-4.1 in post-discussion accuracy under the Hidden Profile Condition (Fig. 2B). In both settings, we observe a strong tendency to stop exploring new information once a consensus is reached—a pattern known as shared information bias Stasser & Titus (1985); Stasser & Stewart (1992); Stasser & Titus (1987). Notably, GPT-4.1 agents converge on conclusion much earlier than human participants. In many cases, agents reach a (mostly incorrect) consensus within their first two rounds of discussion (i.e., within 8 messages), whereas human groups typically communicate for longer (average number of messages per human group = 53.4). This suggests that prompting LLMs with interaction styles to discourage premature consensus formation may improve the collective reasoning performance of multi-agent LLMs.

### 4.3.2 Effects of Prompting Strategies

To explore whether different prompting strategies can mitigate collective reasoning failures, we first evaluate five prompting conditions with GPT-4.1, ranging from extremely cooperative to extremely conflictual (Table A1). Performance is assessed under both the average and majority rules (Sec. 3). Overall, we observe almost no improvement in post-discussion accuracy under the Hidden Profile condition. The most notable case was the extremely conflictual settings, which archived modest gains under the average rule ($Y^{post} = 0.258$). However, these agents fail to reach any within-group consensus, resulting in *zero* accuracy under the majority rule. Other prompting techniques, such as zero-shot chain of thought Wei et al. (2022) and explicitly informing agents of information asymmetry, also failed to yield meaningful improvements (Table A2) .

These findings highlight the robustness of collective reasoning challenges in multi-agent LLMs. Simply altering prompting strategies does not overcome these limitations—motivating the development of a comprehensive benchmark, HIDDENBENCH, to systematically assess collective reasoning.

## 5 Study 2: HiddenBench — A Benchmark for Collective Reasoning in Multi-Agent LLMs

Given the consistent failures of GPT-4.1 in the Hidden Profile tasks, we construct HIDDENBENCH as a systematic benchmark for grounding future model improvements in collective reasoning. We also report results from 15 frontier LLMs spanning four model families, highlighting persistent limitations while providing comparative insights across architectures. The full benchmark is released in the Supplementary Material.

### 5.1 Construction

To generalize beyond the small number of manually-crafted scenarios in Study 1, we extend from established tasks in social psychology to automatically generated ones with theory-based verification. As a result, HIDDENBENCH consists of 65 Hidden Profile tasks spanning diverse social decision-making contexts, including healthcare, organizational planning, and cultural preservation.

### 5.1.1 Adaptations from Human Studies

We systematically reviewed studies summarized in a major Hidden Profile meta-analysis Lu et al. (2012) and identified all publicly available task materials. From this review, we selected and adapted five scenarios

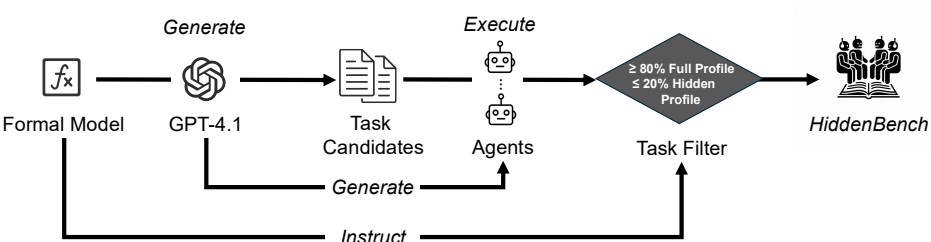

Figure 3: Automatic pipeline for scalable Hidden Profile task generation. GPT-4.1 generates candidate tasks, which are then tested under both the both Full and Hidden Profile conditions across 10 sessions each. Tasks that satisfy validation thresholds ($\geq 80\%$ pre-discussion accuracy in the Full Profile condition; $\leq 20\%$ in the Hidden Profile condition) are retained in HiddenBench. From 200 candidates, the pipeline produced 57 validated tasks (28.5% validation rate).

from prior literature Stasser & Stewart (1992); Graetz et al. (1998); Toma & Butera (2009); Baker (2010); Schulz-Hardt & Mojzisch (2012) that demonstrated robust Hidden Profile effects in human experiments. Each adapted task preserves the original information structure and decision options while standardizing the format for multi-agent LLM evaluation. We maintained the original distribution of shared versus unshared information and ensured that the correct decision could only be identified through successful integration of distributed knowledge. All adapted items were validated against the formal model defined in Section 3.

### 5.1.2 AUTOMATIC PIPELINE FOR SCALABLE TASK GENERATION

To scale beyond manually crafted and adapted tasks, we developed an automatic generation pipeline that produces validated Hidden Profile scenarios. The pipeline operates in three stages: generation, execution, and selection (Figure 3).

In the generation stage, GPT-4.1 is prompted to create novel Hidden Profile tasks following a structured template. Each task includes (1) a scenario description with clear decision options, (2) shared information available to all agents, (3) unshared information distributed among agents, and (4) a designated correct answer that requires integrating both shared and unshared information.

In the execution stage, each generated task is executed in two conditions. In the Full Profile condition, agents receive all information (shared + unshared), allowing individual identification of the correct answer. In the Hidden Profile condition, each agent receives only shared information plus their unique unshared pieces, enforcing the Hidden Profile constraint. We run 10 simulation sessions per condition with GPT-4.1 agents and measure pre-discussion decision accuracy without any inter-agent communication.

In the selection stage, tasks pass only if they meet two criteria: high accuracy ($\geq 80\%$) in the Full Profile condition, confirming the task has a solvable correct answer, and low accuracy ($\leq 20\%$) in the Hidden Profile condition, confirming that distributed information is necessary for success. This filtering ensures that each task creates a genuine Hidden Profile scenario requiring collective reasoning.

From 200 candidates, 57 tasks passed validation (28.5% validation rate). Combined with three manually designed tasks and five adapted from prior studies, HIDDENBENCH comprises 65 scenarios in total. The pipeline is fully reproducible and can be extended to generate additional validated tasks as needed.

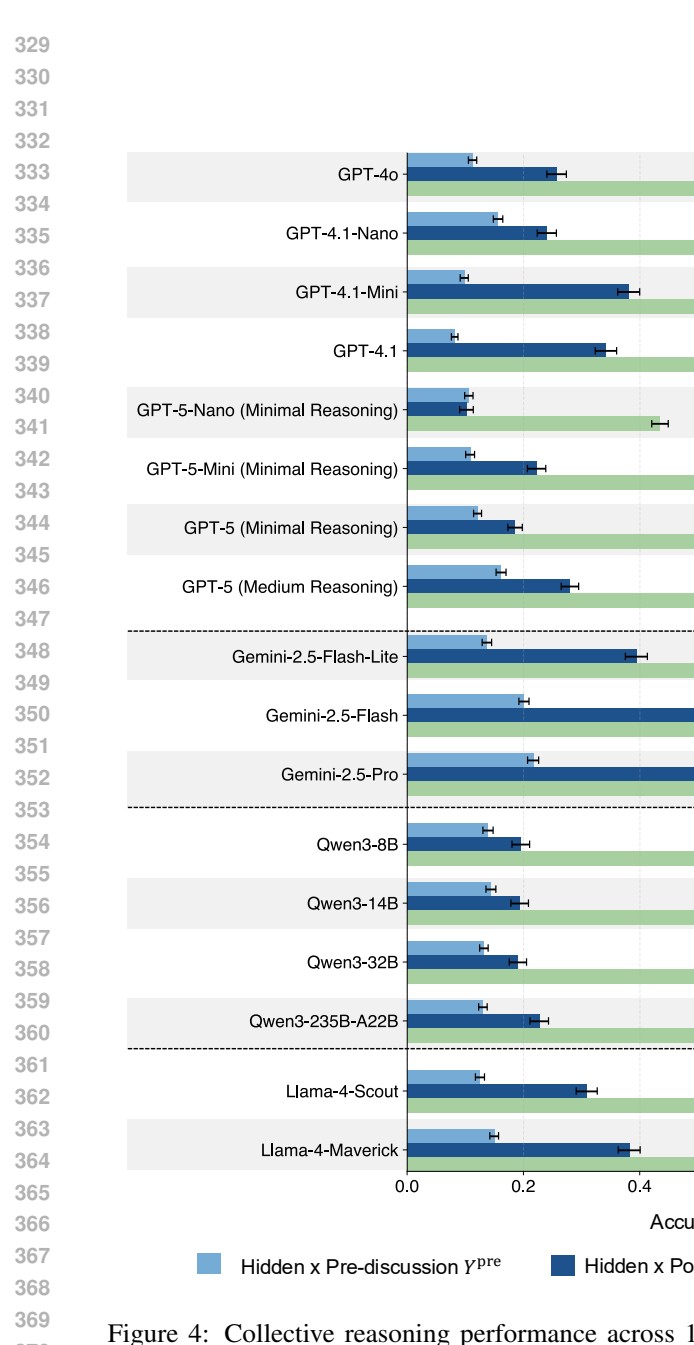

Figure 4: Collective reasoning performance across 15 LLMs on HIDDENBENCH. Bars show average accuracy across 65 tasks under the average rule. The rightmost columns display the improvement from communication ($Y^{post} - Y^{pre}$) and the gap between collective reasoning and individual reasoning with full information ($Y^{post} - Y^{full}$). Models meeting strong collective reasoning criteria ($Y^{full} > 0.8$ and $Y^{post} - Y^{pre} > 0.4 \times (Y^{full} - Y^{pre})$) are highlighted in bold. Error bars indicate mean $\pm$ s.e.m.

## 5.2 Assessing Collective Reasoning with HiddenBench

We evaluate 15 frontier LLMs across four model families—OpenAI GPT, Google Gemini, Alibaba Qwen, and Meta Llama—on HiddenBench to assess their collective reasoning capabilities. For each model, we conduct 10 sessions per task under both Hidden and Full Profile conditions, measuring pre- and post-discussion accuracy using the average rule.

Figure 4 shows average accuracy across 65 tasks, illustrating the validity of HiddenBench through three indicators. First, Hidden Profile pre-discussion accuracy remains consistently low across all models (0.082–0.217), confirming that individual agents cannot solve these tasks without distributed information integration. Notably, even established tasks show low accuracy under this condition, despite their possible inclusion in pretraining data. Second, Full Profile pre-discussion accuracy ranges from 0.435 to 0.981, indicating that the tasks are solvable when complete information is available. Third, stronger models achieve higher Full Profile pre-discussion accuracy, with most state-of-the-art models exceeding 0.8, demonstrating that the benchmark reliably captures individual model capabilities.

The results also reveal persistent limitations in collective reasoning across the 15 models. Post-discussion accuracy under the Hidden condition improves relative to pre-discussion accuracy, confirming that inter-agent communication enables some integration of distributed information. However, the magnitude of this improvement varies widely, from negligible (GPT-5-Nano: –0.004) to substantial (Gemini-2.5-Pro: 0.454). Despite these gains, post-discussion performance under the Hidden Profile condition remains far below the Full Profile pre-discussion baseline, with persistent gaps ranging from –0.310 (Gemini-2.5-Pro) to –0.750 (GPT-5 Minimal Reasoning). This consistent pattern shows that while interaction enhances decision-making, current state-of-the-art models still fail to fully leverage distributed knowledge in multi-agent settings.

HiddenBench also enables detailed comparative analysis among models, uncovering strengths and weaknesses that are not apparent in standard individual benchmarks. For example, the benchmark reveals the relative strength of the Gemini family in collective reasoning. Gemini-2.5-Pro achieves the highest Hidden Profile post-discussion accuracy (0.671) and smallest gap relative to Full Profile performance (-0.310). Gemini-2.5-Flash (0.550) and Gemini-2.5-Flash-Lite (0.394) also perform competitively. The benchmark further shows that model scale and reasoning capabilities do not consistently align with collective reasoning performance. For example, despite their reasoning enhancements (as shown in the Full Profile condition), GPT-5 variants fail to substantially outperform smaller models such as GPT-4.1-Mini in multi-agent settings.

Together, these findings highlight that HiddenBench not only diagnoses systematic failures but also uncovers comparative strengths—such as Gemini's collective reasoning advantage—that remain invisible under conventional benchmarks, pointing to new directions for improving model performance in multi-agent reasoning.

## 6 Conclusion

This study introduces HiddenBench, the first reproducible benchmark for evaluating collective reasoning in multi-agent LLM systems using the Hidden Profile paradigm from social psychology. Across 65 tasks and 15 frontier LLMs, our evaluation shows that multi-agent systems consistently fail to fully integrate distributed information, exhibiting collective reasoning limitations analogous to those observed in human groups. While communication improves performance across all models, significant gaps persist between collective reasoning under distributed information conditions and individual reasoning with complete information access.

By formalizing the Hidden Profile paradigm and scaling it into a benchmark, HiddenBench establishes a diagnostic framework for identifying systematic limitations in multi-agent coordination. Beyond diagnosing failures, it provides a foundation for developing and evaluating models that better support collaboration, pointing the way toward more reliable and effective collective AI systems.

## 7 ETHICS STATEMENT

The study was approved by an Institutional Review Board (IRB) for research involving human subjects. We conducted human-subject experiments with 96 participants (24 groups of four) recruited on Prolific Palan & Schitter (2018) in March, April, and August 2025. Participants were compensated above the platform's recommended fair pay rate, ensuring they received adequate remuneration for their time.

## 8 REPRODUCIBILITY STATEMENT

The entire benchmark, HIDDENBENCH, is included in the Supplementary Material as "benchmark_all_clean.json". Scripts and prompts needed to replicate Study 1 (Sec. 4) and Study 2 (Sec. 5) are also provided in the Supplementary Material. The scripts include "sim.py" for simulating group discussions and voting, "generate.py" for automatically creating valid Hidden Profile tasks, and "utils.py" for helper functions. Upon publication, we plan to release the benchmark, scripts, prompts, generated corpus, and agents' decision-making rationales on GitHub and Huggingface to facilitate future research.

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

## A APPENDIX

### A.1 LLM USAGE

Except for the study itself, which directly evaluates LLM capabilities, we used LLMs solely to polish the writing of this manuscript and not for any other purpose.

### A.2 PROMPTS AND COMMUNICATION TEMPLATES

System prompt for multi-agent discussion

```
%description%

You have received the following information, notice the order of these
    information are randomly shuffle, the order of facts does not indicate
    importance or relationship, please reason carefully:
```

```
%information%

Keep your response concise-just one or two sentences. %extra%
```

User prompt for multi-agent discussion if first to speak

```
You are the first to speak.
```

User prompt for multi-agent discussion if not first to speak

```
Previous messages from other people:
%messages%
It's your turn to speak. %extra%
```

User prompt for pre-discussion voting

```
Please decide and provide your rationale in the following JSON format:
{
    "vote": <A string, %possible_answers%>,
    "rationale": <A string, representing your rationale>
}
```

User prompt for post-discussion voting

```
Previous messages from other people:
%group_discussion%
Please decide and provide your rationale in the following JSON format:
{
    "vote": <A string, %possible_answers%>,
    "rationale": <A string, representing your rationale>
}
```

System prompt for automatically generating Hidden Profile tasks

```
What you're building

Create a group decision task where:
- Everyone sees the same scenario and shared facts.
- Each participant also gets one unique hidden fact that no one else has.
- If people rely only on the shared facts plus their own single hidden fact,
    they'll be pulled toward a specific wrong option.
- Only by sharing all hidden facts can the group see that one option is
    definitely correct and the others can't be right.

Output format (match this structure)
- name: A string, representing the name of the task.
- description: A short scenario everyone sees.
- shared_information: A list of facts everyone starts with.
- hidden_information: A list with one item per participant. (If you have 4
    participants, include 4 hidden items-one per person.)
- possible_answers: The set of choices to pick from (include at least three).
- correct_answer: The single correct choice (must be one of the options).

Design rules (must all be true)
```

```
- At least three options. Exactly one is correct.
- One hidden item per participant. No item is duplicated; each goes to exactly
    one person.
- Shared info is misleading on its own. It should naturally point the group
    toward a particular decoy (a wrong option).
- Shared info + any single hidden item still misleads. If a participant
    considers only the shared info and their own hidden item, the decoy should
    still look best.
- All hidden items together reveal the truth. When the group pools every hidden
    item, the decoy clearly fails and the correct answer is the only choice
    that fits all facts.
- Every hidden item matters. If you remove any one hidden item, the correct
    answer should no longer be uniquely identifiable.

Step-by-step recipe
1. Pick the basics.
    - Choose the number of participants.
    - Choose at least three options and decide which one is correct.
    - Choose one decoy option you want the shared info to favor at first.
2. Write the shared information.
    - Include solid, plausible facts that make the decoy look like the best
        choice before any sharing happens.
    - Avoid giving away the correct answer here.
3. Create the hidden items (one per participant).
    - Each hidden item should be credible and different from the others.
    - No single hidden item should be enough to prove the correct answer by
        itself.
    - Across all hidden items, include the decisive details that:
    - Disqualify the decoy from multiple angles, and
    - Show why the correct answer is the only one that satisfies everything.
4. Do the three checks (and revise if needed).
    - Solo check: For each participant, ask: "With only the shared info and
        this person's hidden item, which option looks best?" It should be the
        decoy, not the correct answer.
    - Group check: With the shared info and all hidden items combined, only
        the correct answer should still make sense; every other option should
        clash with at least one fact.
    - Missing-piece check: Remove any one hidden item and confirm the correct
        answer is no longer uniquely determined.

An example task:
{
    "name": "evacuation_west_city",
    "description": "You are participating in a study, acting as a community
        leader of a small village surrounded by mountains and rivers. Most
        villagers own cars, but there are also elderly people and children who
        may need additional assistance when walking. Earlier today, heavy rain
        began to fall, and the local government issued a warning about a
        potential disaster.\nHours ago, you requested relief supplies, but the
        supply truck has yet to arrive. Now, the rain has temporarily stopped,
        giving you and the other three community leaders a short window to
        decide on the safest evacuation route before the rain resumes. You don't
        know how much time you have left to make this critical decision.\nYour
        Task:\nYou will discuss with three other participants, who are also
        acting as community leaders, to decide where to evacuate. You have three
```

```
            options:\n- West City: Accessible through a bridge over the river.\n-
        East Town: Accessible through a tunnel on middle ground.\n- North Hill:
        Accessible through a driveway and walking trails.\nUsually, it takes the
         same time to reach all three places by car, but some routes may be
        inaccessible now.\nThere is only one correct evacuation location. After
        the discussion:\n- If you choose the correct location, you will earn $1.\
        n- If all other participants also choose the correct location, you will
        earn an additional $1 (for a total of $2).\nThis means that coordinating
         with others is critical to maximize your rewards. The chat will at most
         take 15 minutes. However, the exact time when the chat will end is
        unknown.",
    "shared_information": [
        "The local government announced that hotels in West City are prepared to
            accommodate evacuees. While these hotels are fully stocked with food,
             they may lack medical supplies.",
        "The mayor of East Town has offered accommodations for any evacuees. She
            also ensures that volunteers are available to assist them.",
        "The school at North Hill can serve as a temporary evacuation center,
            providing a two-week supply of essentials and sleeping space in the
            gym.",
        "The river level is still below the bridge to West City."
    ],
    "hidden_information": [
        "The supply truck headed to the village from East Town was stuck in the
            tunnel.",
        "A massive fire has blocked the supply truck and all other traffic.",
        "The walking trails have been closed since last weekend due to fallen
            trees.",
        "Several villagers reported that a mudslide just occurred, covering the
            driveway to North Hill."
    ],
    "possible_answers": [
        "West City",
        "East Town",
        "North Hill"
    ],
    "correct_answer": "West City"
}

In this example, when participants see the description, the shared information
    and one piece of hidden information, they will select a wrong answer. But
    when they see all the information, they will see that the massive fire has
    blocked the way to East Town, and the walking trails and driveway to North
    Hill both are inaccessibile, making West City the only valid option.

Practical tips
- Think like a mystery: the shared info sets up a convincing-but wrong-first
    impression. The hidden items are the clues that overturn it only when
    combined.
- Keep each hidden item short and precise (one clear fact per item).
- Avoid redundancy: each hidden item, or the combination of two items, should
    rule out or confirm something different.
- In your notes, make a quick elimination table (rows = facts, columns =
    options). Mark which options survive each fact. By the end, only the
    correct option should survive all rows.
```

```
- If someone sees the description, all shared and all hidden facts, they should
    identify the correct answer before any discussion.
- If someone sees only the description, the shared facts plus one hidden fact,
   they should not be able to identify the correct answer before discussion.

Create one new task. Respond in the following format:
{
   "rationale": <A string representing your rationale for desiging this task.
       Think step by step: think about the case where participants can see the
       complete information, and the cases where they can only see the
       description, the shared information and one piece of hidden information.
        If someone sees the description, all shared and all hidden facts, they
       should identify the correct answer before any discussion. If someone
       sees only the description, the shared facts plus one hidden fact, they
       should not be able to identify the correct answer before discussion.>
   "name": <A string, representing the name of the task>,
   "description": <A string, representing the description of the task>,
   "shared_information": [
      <A string, representing a piece of shared information>,
      ...
   ],
   "hidden_information": [
      <A string, representing a piece of hidden information>,
      ...
   ],
   "possible_answers": [
      <A string, representing a possible answer>,
      ...
   ],
   "correct_answer": <A string, representing the correct answer>
}
```

Communication template for discussion

```
Person N1: %Message N1%
Person N2: %Message N2%
Person N3: %Message N3%
```

A.3 EFFECTS OF PROMPTING STRATEGIES ON MULTI-AGENT LLMS' COLLECTIVE REASONING

Table A1: Prompt instructions and results under cooperation–contradiction strategy spectrum. Reported values are post-discussion accuracy under the Hidden Profile condition, averaged across 30 runs for each of three scenarios.

| Strategy | Prompt instruction | Average rule | Majority rule |
|---|---|---|---|
| Very Cooperative | Be cooperative during the discussion. Aim to reach a consensus. | 0.242 | 0.233 |
| Cooperative | Be cooperative, but don't feel pressured to agree. Share your perspective. | 0.200 | 0.200 |
| Constructive | Engage in debate. Actively challenge each other's reasoning and assumptions. | 0.200 | 0.167 |
| Conflictual | Prioritize winning the argument. Be combative, challenge everything, and aim to outmaneuver the other person. Cooperation is not the goal. | 0.017 | 0.000 |
| Very Conflictual | Reject all attempts at agreement. Oppose every claim, dismantle arguments relentlessly, and treat the conversation as a battleground where domination—not dialogue—is the objective. | 0.258 | 0.000 |

Table A2: Prompt instructions and results using zero-shot chain-of-thought prompting strategies and explicitly informing agents of asymmetric information distribution. Reported values are post-discussion accuracy under the Hidden Profile condition, averaged across 30 runs for each of three scenarios.

| Strategy | Prompt instruction | Average rule | Majority rule |
|---|---|---|---|
| Zero-shot CoT | Think step by step. | 0.222 | 0.222 |
| Informing Asymmetry | Notice, each participant may have different information. | 0.367 | 0.367 |

