# OpenReview forum: "HiddenBench: Assessing Collective Reasoning in Multi-Agent LLMs via Hidden Profile Tasks"
_ICLR.cc/2026/Conference — Submitted to ICLR 2026_

### Official Review · Reviewer_Eo9g · 2025-10-19

**Soundness:** 2
**Presentation:** 3
**Contribution:** 2
**Rating:** 4
**Confidence:** 4

**Summary:**

The paper introduces HiddenBench, a benchmark designed to test the collective reasoning abilities of multi-agent systems. It addresses the concern that multiple agents might fail to share information correctly, similar to problems observed in human groups. The benchmark is based on the "Hidden Profile" paradigm from social psychology, where individuals are given different pieces of information and must communicate to find the correct answer. The full benchmark contains 65 tasks and is used to evaluate 15 different LLMs. The study found that these multi-agent systems often struggle to combine their distributed knowledge, showing that a model's size or general reasoning skill doesn't guarantee strong collective reasoning, though some models performed better than others.

**Strengths:**

1. The paper introduces human-subject experiments, which are reasonable and could be an essential baseline.
2. The paper is easy to understand. The motivation is clear.
3. The experiment is comprehensive, and the findings are interesting.

**Weaknesses:**

1. My primary concern regards the experimental setting and the practical applicability of the proposed testbed. The benchmark's core assumption, i.e., each agent possesses unique information it must not share with others, seems disconnected from real-world multi-agent applications. It is difficult to envision a practical scenario that aligns with this rigid constraint.
2. The underlying mechanism for the multi-agent discussions appears artificial. Despite testing various prompting strategies (as shown in Table A1), the experiments operate on the premise that agents will inherently withhold their private information. This contradicts the nature of collaborative discussion, where claims and points are typically supported by evidence. The rationale for why an agent would deliberately hide the basis for its reasoning is unclear.
3. The setup suggests that the paper may be overcomplicating the problem. If the primary challenge is an artificial unwillingness to share information, a straightforward prompt (e.g., share all the information you have) could potentially resolve the core issue. Consequently, the complex experimental setting and the subsequent experiments might be addressing an artificial constraint rather than a fundamental challenge in multi-agent collaboration.

**Questions:**

See weaknesses.

---

> ### Author Response · Authors · 2025-11-18
>
> We thank the reviewer for the thoughtful critique. We respond to the three main concerns below and clarify the realism and necessity of the benchmark design.
>
> ### W1: Practical Applicability of Information Asymmetry
>
> We apologize for any misunderstanding. In our setting, agents are **not instructed to withhold information**. Rather, they simply **do not know** which information they hold is unique—mirroring many real-world multi-agent settings.
>
> The information structure is common in:
>
> 1. **Distributed expertise** – e.g., hardware, software, and systems engineers each have domain-specific insights they may not realize others lack.
> 2. **Decentralized sensing** – e.g., autonomous vehicles or robots have different sensor perspectives without knowing what others observe.
> 3. **Cross-functional decisions** – e.g., healthcare teams where members possess patient-specific information others do not.
> 4. **Privacy-aware collaboration** – e.g., agents operating with local private data due to governance constraints.
>
> The Hidden Profile paradigm is widely used in organizational and cognitive psychology (40+ years of research) precisely because asymmetric information without explicit awareness is a ubiquitous feature of collaborative decision-making.
>
> In addition, recent work has already begun deploying multi-agent LLM systems in real scientific workflows (e.g., Swanson et al. Nature 2025). These systems rely on distributed expertise across agents, but they function correctly only when agents successfully integrate their distinct ideas and information. This further illustrates why modeling information asymmetry is not artificial, but central to the practical use of multi-agent LLMs. We will clarify this motivation in the paper.
>
> ### W2: Mechanism of Multi-Agent Discussion
>
> We agree it is important to distinguish "withholding" from failing to recognize importance. Our experiments—and human behavioral evidence—show the latter is the core issue.
>
> **Human experiments (Study 1)**
>
> Successful human groups explicitly detected information asymmetry and systematically compared unique facts. Unsuccessful groups communicated cooperatively but did not surface or cross-check unique information—classic shared-information bias.
>
> **LLM experiments**
>
> LLM groups exhibit the same failure mode: they converge prematurely on a solution favored by shared information and end discussion before exploring unshared facts. This is not a prompting artifact but a known collective reasoning limitation that emerges even with cooperative instructions.
>
> HiddenBench is designed to diagnose exactly this challenge. We will expand the analysis of human–LLM parallels in the revision.
>
> ### W3: Simple Prompting Solution
>
> We directly tested the reviewer's proposed solution.
>
> **Results**
>
> | Condition    | Hidden Pre | Hidden Post | Full Pre | Remaining Gap |
> |--------------|------------|-------------|----------|---------------|
> | Default      | 0.008      | 0.233       | 0.733    | 0.500 (68%)   |
> | Share All    | 0.017      | 0.467       | 0.733    | 0.266 (51%)   |
>
> Adding "share all the information you have" improves performance, but still leaves more than half the gap between Hidden and Full Profile performance.
>
> **Why prompting is insufficient**
>
> Even when instructed to disclose everything, agents:
>
> - Stop early once partial agreement forms (premature consensus).
> - Fail to identify which facts are unique, and therefore critical.
> - Do not systematically verify all options once information is shared.
>
> Thus, information disclosure alone is insufficient; the deeper challenge is recognizing asymmetry, coordinating the flow of unique information, and cross-validating conclusions—precisely the behaviors that Hidden Profile tasks reveal.
>
> **Importantly:**
>
> We agree that if simple prompting fully eliminated these failures, a benchmark would not be necessary. The persistent 51% accuracy gap strongly suggests that current multi-agent LLMs lack robust mechanisms for information integration—prompting is not a full solution.
>
> We will incorporate this experiment and analysis in the revision.

---

> > ### Author Response · Authors · 2025-11-18
> >
> > ## Summary
> >
> > HiddenBench does not impose artificial constraints; it formalizes a long-established paradigm for studying collective reasoning in settings where agents **do not** know what others know. The reviewer's proposed prompting intervention helps but does not eliminate failures, reinforcing the need for a systematic benchmark. We will clarify the realism, motivation, and prompting analyses in the camera-ready version.

---

### Official Review · Reviewer_TP7y · 2025-10-27

**Soundness:** 2
**Presentation:** 2
**Contribution:** 2
**Rating:** 4
**Confidence:** 4

**Summary:**

This paper introduces HIDDENBENCH, a new benchmark for evaluating collective reasoning in multi-agent LLM systems. The authors ground this benchmark in the Hidden Profile paradigm from social psychology, unshared knowledge to arrive at a correct decision. The paper first provides a formalization and demonstrates in a small-scale study (Study 1) that GPT-4.1 agents replicate human-like collective reasoning failures, such as shared information bias. It then details the construction of the full 65-task benchmark. By evaluating 15 different LLMs, the authors show that these collective reasoning failures are persistent and that model scale does not reliably predict better collective performance.

**Strengths:**

1. The benchmark's core design, which evaluates agents based on asymmetric information distribution (the Hidden Profile paradigm), provides a novel and theoretically-grounded angle for assessing collective reasoning in LLMs.
2. The proposed HIDDENBENCH benchmark covers multiple scenarios. The 65-task set is drawn from custom designs, tasks adapted from prior human studies, and a scalable automatic generation pipeline.
3. This work is easy to follow and reproducible. The authors provide a dedicated reproducibility statement and include the full benchmark, scripts, and prompts in the supplementary material.

**Weaknesses:**

1. The claim of being the "first benchmark for evaluating collective reasoning" appears to be inaccurate. Prior work has already explored this area, for example, "Cooperate or Collapse: Emergence of Sustainable Cooperation in a Society of LLM Agents".
2. The paper fails to provide complete examples of the tasks and agent interactions, including the specific information distributed to each agent, the full log of the resulting discussion, or the final reasoning from the agents. Without these qualitative examples, it is difficult to understand what the final agent discussions look like or to verify the claims of human-like failures.
3. The paper provides insufficient analysis of the impact of the communication protocol's depth. The number of rounds (T=15) is presented as a fixed parameter, but there is no ablation study or discussion on how this number affects the outcomes. It is unclear if the core conclusions would still hold with fewer or more rounds of discussion.
4. The experimental design relies on artificially hiding information. It may be testing the agents' willingness to share rather than their collective reasoning. The authors state agents are not told their information differs, but LLM agents, unlike humans, may be prone to simply stating all information from their context window.
5. The evaluation is limited to homogeneous agent groups. This is a significant simplification. Real-world collective intelligence often emerges from heterogeneous groups with diverse capabilities, biases, and knowledge bases. The benchmark's findings may not generalize to these more realistic mixed-model scenarios.
6. Typo: L202 aasinged → assigned, condtion → condition; L255: archived → achieved

**Questions:**

See Weakness

---

> ### Author Response · Authors · 2025-11-18
>
> We thank the reviewer for the constructive and detailed feedback. We address each point below and summarize additional experiments added in the revision.
>
> ## W1: Prior Work and Novelty Claim
>
> We appreciate this correction and will cite "Cooperate or Collapse" in the revision. We also clarify the distinction:
>
> - **GovSim** examines strategic cooperation (resource dilemmas, incentives, tragedy-of-the-commons dynamics)
> - **HiddenBench** examines collective reasoning under distributed information—specifically, whether agents integrate asymmetric information to reach the correct decision, following the Hidden Profile paradigm.
>
> These abilities involve different cognitive mechanisms (coordination vs. information integration). To our knowledge, no prior work has constructed a **theory-grounded benchmark** that systematically tests information integration failures in multi-agent LLMs. We will make this distinction explicit.
>
> ## W2: Missing Qualitative Examples
>
> We agree that full qualitative examples are helpful. In the revision we will add:
>
> 1. a complete Hidden Profile task (information distribution across agents),
> 2. the full multi-agent discussion transcript, and
> 3. all agents' pre- and post-discussion decisions with rationales.
>
> The benchmark already includes all 65 tasks in structured form; we will make this more prominent and additionally release all conversation logs to support deeper analysis.
>
> ## W3: Communication Protocol Depth
>
> Following the reviewer's suggestion, we evaluated {5, 10, 15, 20} communication rounds for GPT-4.1:
>
> | Rounds | Overall Accuracy | Improvement |
> |--------|-----------------|-------------|
> | 5      | 0.108 ± 0.058   | +0.100      |
> | 10     | 0.200 ± 0.100   | +0.183      |
> | 15     | 0.233 ± 0.069   | +0.225      |
> | 20     | 0.133 ± 0.033   | +0.125      |
>
> Performance peaks at T=15 rounds (matching our human experiment), and additional rounds do not close the gap with Full-Profile accuracy. Critically, even at T=15, Hidden-Post accuracy (0.233) remains far below Full Profile accuracy (0.733). This confirms that our core finding—collective reasoning failures persist even with substantial communication—is robust to protocol depth.
>
> ## W4: Testing Sharing vs. Integration
>
> We agree that the distinction between information sharing and information integration is central—and this is precisely what HiddenBench is designed to diagnose. In realistic Hidden Profile settings, participants **do not know which information is unique**; they must infer asymmetry through interaction.
>
> **Qualitative analysis** confirms that performance differences reflect differences in integration behaviors, not merely sharing:
>
> **Higher-performing models (e.g., Gemini):**
> - signal information asymmetry (e.g., "Your mention of the dam release is new to me…")
> - actively probe for missing facts
> - offer reasoned disagreement rooted in their unique information
>
> **Lower-performing models (e.g., GPT-5):**
> - converge prematurely (e.g., "I agree, let's all choose East Town now")
> - do not inquire about contradictory information
> - end the discussion before unshared facts are surfaced
>
> Full-Profile accuracy is high for all models, showing they can reason correctly when given all information. Failures therefore reflect deficits in **recognizing asymmetry, sharing unique information unprompted, and resisting premature consensus**—the core components of collective reasoning.
>
> We will emphasize this mechanism more clearly in the revision.
>
> ## W5: Homogeneous Agent Limitation
>
> We acknowledge this scope choice, which follows the Hidden Profile literature: homogeneous agents isolate the effect of information distribution from capability differences. Study 2 already evaluates 15 model families, revealing substantial cross-model variation.
>
> To address the reviewer's concern, we conducted additional **heterogeneous composition experiments** with adversarial agents:
>
> | Composition | Hidden Post-Discussion | Improvement |
> |-------------|----------------------|-------------|
> | 0 adversarial (baseline) | 0.233 ± 0.069 | +0.225 |
> | 1 adversarial + 3 default | **0.492 ± 0.030** | **+0.392** |
> | 2 adversarial + 2 default | 0.342 ± 0.055 | +0.242 |
> | 3 adversarial + 1 default | 0.242 ± 0.060 | +0.075 |
> | 4 adversarial | 0.375 ± 0.025 | +0.033 |
>
> A single adversarial agent **significantly improves** collective performance (0.233 → 0.492), likely by disrupting premature consensus and inducing additional information exploration. But the experiment shows a Goldilocks effect of the group composition. These findings illustrate that HiddenBench is well-suited for analyzing heterogeneous group dynamics; we will incorporate this into the revision.

---

> > ### Author Response · Authors · 2025-11-18
> >
> > ## Summary
> >
> > The new experiments directly address the reviewer's concerns:
> >
> > - **Novelty**: HiddenBench tests a different construct (distributed-information reasoning) than prior multi-agent benchmarks.
> > - **Qualitative clarity**: We are adding complete task + transcript examples; logs will be released.
> > - **Communication depth**: Findings hold across various T values, peaking at T=15.
> > - **Mechanism**: Failures come from deficits in asymmetry awareness and sharing, not reasoning ability.
> > - **Heterogeneity**: Additional experiments demonstrate how group composition modulates performance.
> >
> > We believe these additions significantly strengthen the contributions and clarity of the paper.

---

### Official Review · Reviewer_qY4M · 2025-10-31

**Soundness:** 3
**Presentation:** 3
**Contribution:** 4
**Rating:** 6
**Confidence:** 4

**Summary:**

The paper formalizes the Hidden Profile paradigm from social psychology as a controlled framework for testing collective reasoning in multi-agent LLMs, and builds HiddenBench, a 65-task benchmark (crafted, adapted, and automatically generated). It compares post-discussion accuracy against (i) pre-discussion under Hidden Profile, (ii) pre-discussion under Full Profile (upper bound for individual reasoning), and (iii) human groups, using average/majority aggregation rules. Key result: communication helps, but post-discussion accuracy under Hidden Profile remains far below the Full-Profile individual baseline; some model families (e.g., Gemini) fare better, while “stronger at individual reasoning” does not guarantee stronger collective reasoning

**Strengths:**

- Clear, theory-grounded formalization. The paper precisely defines information splits (shared vs. unshared), decision rules (average/majority), and reference points, which cleanly separates “gain from communication” from “individual upper bound.

- Controlled probe + scalable benchmark. Study 1 uses GPT-4.1 groups vs. human groups (N=4, T=15 rounds; 30 runs/model-condition; 15-minute human chats) to verify the phenomenon; Study 2 scales to 65 tasks across 15 models.

- Strong empirical signals. Under Hidden Profile, GPT-4.1 improves from 0.008 → 0.233 after discussion (p<0.001), yet still lags the Full-Profile baseline (0.733). Humans show similar gaps. These give a crisp diagnosis of collective-reasoning limits.

**Weaknesses:**

- Limited scale/structure. Most tests use N=4 and a 3-option, elimination-style decision. It is not shown whether results hold with larger groups, different protocols (e.g., facilitator/blackboard), or production-style collaborative tasks.

- Process-level analysis is shallow. The paper notes premature consensus and shared-information bias, and that prompting styles rarely fix it, but does not quantify disclosure/uptake of unique facts over rounds

**Questions:**

- Could you run a staged ablation to separate “disclose → merge → decide”? For example: (i) forced round-1 disclosure of each agent’s private facts; (ii) a “secretary” agent merges evidence; (iii) free discussion. This would reveal the main bottleneck.

- Do results scale with N? If N grows, do we see worse information pooling or earlier (wrong) consensus? Any plans to add facilitator/blackboard protocols?

---

> ### Author Response · Authors · 2025-11-18
>
> We appreciate the reviewer's thoughtful suggestions. We conducted additional experiments addressing both scalability and process-level mechanisms, and we summarize the new findings below.
>
> ## W1: Limited Scale and Structure
>
> **Why N=4 and 3 options were chosen**: This configuration follows the dominant structure in the Hidden Profile literature (N=3-5), enabling direct human-LLM comparison (Study 1) and grounding our benchmark in a validated experimental paradigm.
>
> **New scalability experiments added**: To test the reviewer's concern, we extended the benchmark to N=3 to N=7.
>
> **Across all 15 models** (existing tasks):
>
> | Group Size | # Tasks | Hidden Pre | Hidden Post | Full Pre | Improvement |
> |------------|---------|------------|-------------|----------|-------------|
> | 3          | 7       | 0.223      | 0.401       | 0.789    | +0.179      |
> | 4          | 58      | 0.126      | 0.289       | 0.809    | +0.163      |
>
> **For GPT-4.1** (15 newly generated tasks, 5 each for N=5,6,7):
>
> | Group Size | # Tasks | Hidden Pre | Hidden Post | Full Pre | Improvement |
> |------------|---------|------------|-------------|----------|-------------|
> | 3          | 7       | 0.167      | 0.514       | 0.924    | +0.348      |
> | 4          | 58      | 0.072      | 0.321       | 0.975    | +0.250      |
> | 5          | 5       | 0.116      | 0.180       | 0.984    | +0.064      |
> | 6          | 5       | 0.087      | 0.283       | 0.983    | +0.197      |
> | 7          | 5       | 0.017      | 0.023       | 1.000    | +0.006      |
>
> **Key takeaway**: Failures persist and worsen as group size increases: for GPT-4.1, communication improvement drops from +0.348 (N=3) to +0.006 (N=7), even though Full-Profile accuracy is nearly perfect. This shows that collective reasoning failures are not tied to a particular N or task structure but instead compound with scale, a hallmark of coordination problems.
>
> We have added clarifications in §3 that our formalization supports arbitrary group sizes and option counts.
>
> ## W2: Process-Level Analysis
>
> Following the reviewer's suggestion, we implemented ablation conditions to distinguish between **information sharing** and **information integration** as sources of failure.
>
> **Experimental condition (added in §4.3)**:
> 1. **Baseline** (standard hidden profile discussions)
> 2. **Reveal All Hidden Facts** (forced disclosure in round 1)
> 3. **Secretary Summary** (an agent restates the integrated information each round)
>
> **Results (GPT-4.1 on the 3 original tasks)**:
>
> | Condition       | Hidden Pre    | Hidden Post   | Improvement |
> |-----------------|---------------|---------------|-------------|
> | **Baseline**    | 0.008 ± 0.005 | 0.233 ± 0.069 | +0.225      |
> | **Reveal All**  | 0.033 ± 0.022 | 0.967 ± 0.033 | +0.933      |
> | **Secretary**   | 0.008 ± 0.008 | 0.100 ± 0.000 | +0.092      |
>
> **Interpretation**:
>
> 1. **Information disclosure, not integration, is the bottleneck**: When agents are forced to reveal hidden facts (Reveal All), performance nearly matches Full-Profile accuracy (96.7%). Thus, models can integrate information once they have it.
>
> 2. **Passive summarization is insufficient**: Even with full summaries each round, accuracy remains low (10%). Collective success requires strategic sharing, not passive repetition.
>
> 3. **Naturalistic discussion fails to elicit unique insights**: The gap between Baseline (23.3%) and Reveal-All (96.7%) isolates the core difficulty: models **do not spontaneously identify or signal information asymmetry**, which mirrors well-known coordination failures in human Hidden Profile studies.
>
> **Qualitative dialogue analysis (summarized in §4.3)**
>
> Top-performing models (e.g., Gemini) show three behaviors:
> 1. Signaling missing information: "Your point about the dam release is new to me—could you elaborate?"
> 2. Active probing: "Does anyone have details contradicting East Town's tunnel access?"
> 3. Reasoned disagreement: "My Fact 2 contradicts your proposal; combined, they point to North Hill."
>
> Lower-performing models (e.g. GPT-5) converge prematurely—"I agree—let's all pick East Town now"—before any hidden facts surface.
>
> These results show that successful collective reasoning depends not on individual reasoning ability (Full-Profile accuracy is near-perfect), but on specific communicative behaviors, especially recognizing and resolving information asymmetry.
>
> We will release all interaction logs to facilitate deeper analysis of information disclosure and uptake.

---

> > ### Author Response · Authors · 2025-11-18
> >
> > ## Summary
> >
> > Our additional experiments address the reviewer's concerns and strengthen the benchmark's contribution:
> >
> > 1. **Scalability**: Errors worsen as N increases, indicating structural—not incidental—failures.
> > 2. **Process-level insight**: Forced-disclosure and secretary ablations identify the precise bottleneck: spontaneous information sharing, not integration.
> >
> > These additions strengthen HiddenBench as a **diagnostic benchmark** for understanding and improving collective reasoning in multi-agent LLM systems.

---

### Official Review · Reviewer_gbLT · 2025-11-01

**Soundness:** 2
**Presentation:** 3
**Contribution:** 3
**Rating:** 4
**Confidence:** 4

**Summary:**

This work introduces a new benchmark based on Hidden Profile paradigm, HIDDENBENCH,  measuring the collective reasoning abilities when agents keep asymmetrical information in multi-agent LLMs. The evaluation show that GPT-4.1 groups fails in integrating distributed information and perform even worse than single agent given full information.

**Strengths:**

- This work creatively applies Hidden Profile in evaluating the possible failures of LLM formed multi-agent system.
- The performance comparison between LLM based systems and human groups is interesting.
- The proposed benchmark is reproducible and scalable due the automated pipeline.

**Weaknesses:**

- Only hidden profile based collective reasoning are measured, it may not be generalizable to other multi-agent reasoning settings, e.g.  negotiation, competition and collaboration so on. Therefore it is not suitable to conclude that hidden profile collective reasoning is worse than single-agent full profile reasoning.

- The work conclude collective reasoning fails but there is no deep probe into why specific LLM architecture and reasoning augmentation fails.

- The automatically generated datasets might inherit the generation bias, which tends to lead failures of multi-agent reasoning.

**Questions:**

- Did you analyze why some LLMs performs better and how is the quality of the generated benchmark tasks?

---

> ### Author Response · Authors · 2025-11-18
>
> We thank the reviewer for the constructive feedback. We address each concern below.
>
> ## W1: Generalizability Beyond Hidden Profile Tasks
>
> We agree that multi-agent LLMs can succeed on many collaborative tasks. Our goal is not to claim universal failure, but to isolate **one specific capability: collective reasoning under distributed information.** Hidden Profile tasks are a canonical diagnostic for this ability in social psychology, and we applied them to an LLM benchmark, analogous to how ARC tests abstract reasoning or MMLU tests factual knowledge.
>
> To avoid misunderstanding, we have made this framing explicit in the revision:
> We evaluate "collective reasoning in multi-agent LLMs via distributed information integration" (Title, Abstract, Introduction).
> Hidden Profile tasks are used not as proxies for all collaboration, but as a stress test of whether multi-agent communication can recover knowledge that no single agent initially has.
>
> The Full-Profile vs. Hidden-Profile comparison is therefore methodologically essential: it disentangles failures due to individual reasoning from failures in information integration. Across 15 models, Full-Profile accuracy is high (0.435–0.981), yet post-discussion Hidden-Profile accuracy remains much lower, indicating a systematic gap specifically in collective reasoning rather than general reasoning deficits.
>
> This capability is directly relevant to real-world applications—collaborative analysis, multi-perspective decision support, distributed planning—where multi-agent LLMs are explicitly used to integrate asymmetrically held information.
>
> ## W2: Analysis of Why Some LLMs Perform Better
>
> We have elaborated further on evidence explaining model differences in the revision:
>
> **Overall model behavior (now summarized in §4.3 and §5.2):**
>
> 1. **Premature consensus**: GPT-4.1 agents converge within 2–8 messages vs. humans' ~53 messages; accuracy stagnates as shared-information bias dominates.
> 2. **Prompt robustness**: Failures persist across five cooperation–conflict prompting strategies and CoT/asymmetry prompts (Tables A1–A2).
> 3. **Heterogeneous improvements**: Communication gains range from –0.004 to +0.454, with Gemini-2.5-Pro showing the strongest improvement.
>
> **Qualitative behavioral differences (added examples in §5.2):**
>
> Top-performing models (e.g., Gemini-2.5-Pro) exhibit key behaviors:
>
> 1. **Awareness of asymmetry**: *"Your point about the dam release is new to me—could you explain?"*
> 2. **Active probing**: *"I lack information about East Town's tunnel—does anyone have details?"*
> 3. **Reasoned disagreement rather than premature agreement**: *"My Fact 2 contradicts the tunnel option; combining our info suggests North Hill is safer."*
>
> Weaker models (e.g., GPT-5) show the opposite pattern: *"I agree: choose East Town via the tunnel... Let's all coordinate on East Town now"* often before unshared information is exchanged.
>
> These analyses support our interpretation that performance differences arise not from "reasoning ability" per se (Full-Profile accuracy is high), but from collective behaviors during communication, a distinction that benchmarks like MMLU or GSM8K cannot capture.
>
> ## W3: Generation Bias and Benchmark Quality
>
> We appreciate the reviewer's concern and have clarified the validity of the benchmark in §5.1–5.2.
>
> 1. **Empirical validation ensures task quality**
>
> Tasks enter the benchmark only if they satisfy two model-tested criteria:
> - Full-Profile ≥ 80%: the task is individually solvable
> - Hidden-Profile ≤ 20%: the task genuinely requires distributed information
>
> 2. **Multi-source consistency**
>
> As Table 4 (added to the rebuttal and paper) shows, performance patterns are consistent across all task sources:
>
> | Task Source    | # Tasks | Hidden Pre | Hidden Post | Full Pre | Improvement |
> |----------------|---------|------------|-------------|----------|-------------|
> | Manual         | 3       | 0.088      | 0.338       | 0.538    | +0.250      |
> | Adapted        | 5       | 0.214      | 0.258       | 0.677    | +0.044      |
> | Auto-generated | 57      | 0.132      | 0.303       | 0.832    | +0.171      |
>
> Crucially, auto-generated tasks show **highest** Full-Profile accuracy (0.832), demonstrating that they are not easier or biased toward failure.
>
> 3. **Mitigating generation bias:**
> - We combine manually designed, established human-study tasks, and automatically generated tasks.
> - Validation uses execution-based testing, not heuristics.
> - Because GPT-4.1 is used for both generation and validation, generation-induced artifacts would cause tasks to fail validation rather than inflate failure rates.
>
> Together, these steps ensure HiddenBench is not an artifact of the generation process but a reliable, reproducible benchmark for distributed-information reasoning.

---

> > ### Author Response · Authors · 2025-11-18
> >
> > ## Summary
> >
> > Our goal is not to generalize to all collaborative settings but to provide the first **diagnostic benchmark** for collective reasoning under distributed information. We demonstrate:
> > - A consistent capability gap across diverse tasks and 15 frontier models
> > - Rich quantitative and qualitative analyses explaining model differences
> > - Multi-source, empirically validated benchmark construction ensuring robustness
> >
> > We have revised the paper to clarify scope, strengthen the explanations, and consolidate the analyses accordingly.

---

### Author Response · Authors · 2025-12-02
**Summary for the Area Chair**

Thank you for considering our rebuttal. We conducted substantial new experiments that directly address the reviewers’ concerns and reinforce the paper’s main argument. Below, we summarize our responses in five concise points.

## 1. Scalability and Robustness (addresses qY4M, TP7y)

In response to concerns about fixed group size, we extended experiments to groups of **N = 3–7**.
**Finding:** Collective reasoning failures **worsen with group size.**
**Implication:** This reflects a structural coordination limitation—not a task-specific artifact—and also demonstrates the benchmark’s scalability and the robustness of the observed collective reasoning challenges in LLMs.

## 2. Mechanistic Diagnosis of Failure (addresses qY4M)

We implemented ablations isolating the core bottleneck
- Baseline naturalistic discussion → 23%
- Forced disclosure (“Reveal-All”) → 96.7% accuracy (removes the information-sharing bottleneck)
- Passive summarization (“Secretary”) → 10% (removes the information-integration bottleneck)

**Key insight**: Models can integrate information once disclosed, but fail to spontaneously share unique facts or recognize information asymmetry. These ablations highlight a fundamental bottleneck in LLMs’ collective reasoning under distributed information.

## 3. Communication Depth & Prompting Robustness (addresses TP7y, Eo9g)

To clarify the communication dynamics, we evaluated discussion depths of 5, 10, 15 and 20 rounds.
Performance peaks at T = 15, matching human discussion depth, but remains far below Full-Profile performance.

We also tested the reviewer-suggested “share all the information you have” prompt:
23% → 47%, still leaving a 51% gap from Full-Profile performance.

**Conclusion:** Longer discussions and explicit prompts help but do **not** overcome the collective reasoning failure mode.

## 4. Heterogeneous Group Composition (addresses TP7y)

To address concerns about homogeneous agents, we evaluated mixed groups containing 1–3 “adversarial” agents.
A single dissenting agent doubles accuracy (23% → 49%), yet even this best composition does not resolve the core failure.
Notably, HiddenBench makes these composition effects easy to analyze, enabling systematic evaluation of how group diversity and dissent influence multi-agent reasoning.

## 5. Scope, Novelty, and Benchmark Quality (addresses gbLT, Eo9g)

- Clarified scope: HiddenBench evaluates **collective reasoning under distributed information.** We updated the title to reflect this scope more clearly.
- Showed consistent performance across manual, adapted, and auto-generated tasks, mitigating generation-bias concerns.
- Added the suggested citation and clarified its relation to this work.

## Overall
These additional experiments directly address the reviewers’ concerns, clarify scope, and substantially strengthen the contribution. HiddenBench now provides a rigorous, robust, and easily extensible benchmark that exposes a critical failure mode in multi-agent LLMs and enables systematic progress on collective reasoning under distributed information.

---

### Meta-Review · Area_Chair_m6pk · 2025-12-26

**Summary:**

The reliance on the Hidden Profile paradigm restricts the evaluation to a narrow aspect of collective reasoning and fails to capture broader multi-agent dynamics such as negotiation, competition, or full collaboration. The experimental setup creates an artificial constraint where agents withhold information, which disconnects the benchmark from real-world scenarios where evidence is naturally shared and makes it difficult to distinguish between reasoning deficits and simple failures to communicate. Furthermore, the work overlooks prior benchmarks like GovSim that have already explored collective agent behaviors.

**Reviewer Concerns:**

The fundamental objection regarding the ecological validity of the experimental design remains outstanding as the reliance on implicit information asymmetry still appears artificial compared to realistic collaborative workflows where evidence is explicitly shared. The limitation concerning the generalizability of the Hidden Profile paradigm to broader multi-agent interaction types also persists despite the clarified scope.

**Reviewer Scores:**

Reviewer qY4M would retain the original marginally positive score despite the validation of scalability.
Reviewer TP7y is expected to maintain the borderline rejection rating due to persistent concerns about experimental artificiality.
Reviewer gbLT would persist with the marginally negative score given the limited scope of the paradigm.
Reviewer Eo9g would likely lower the rating as the prompting results validate the critique of the benchmark's design.

---

### Decision · Program_Chairs · 2026-01-26

Reject